# Preparation, Characterization, Wound Healing, and Cytotoxicity Assay of PEGylated Nanophytosomes Loaded with 6-Gingerol

**DOI:** 10.3390/nu14235170

**Published:** 2022-12-05

**Authors:** Ali Al-Samydai, Moath Al Qaraleh, Walhan Alshaer, Lidia K. Al-Halaseh, Reem Issa, Fatima Alshaikh, Aseel Abu-Rumman, Hayat Al-Ali, Emad A. S. Al-Dujaili

**Affiliations:** 1Pharmacological and Diagnostic Research Centre, Department Pharmacological, Faculty of Pharmacy, Al-Ahliyya Amman University, Amman 19328, Jordan; 2Cell Therapy Center, The University of Jordan, Amman 11942, Jordan; 3Department of Pharmaceutical Chemistry, Faculty of Pharmacy, Mutah University, Al-Karak 61710, Jordan; 4Pharmacological and Diagnostic Research Center, Faculty of Allied Medical Sciences, Al-Ahliyya Amman University, Al-Salt 19328, Jordan; 5Centre for Cardiovascular Science, Queen’s Medical Research Institute, University of Edinburgh, 47 Little France Crescent, Edinburgh EH16 4TJ, UK

**Keywords:** cytotoxicity, 6-gingerol, liposomes, nanoparticles, phytosomes, wound healing

## Abstract

Background: Nutrients are widely used for treating illnesses in traditional medicine. Ginger has long been used in folk medicine to treat motion sickness and other minor health disorders. Chronic non-healing wounds might elicit an inflammation response and cancerous mutation. Few clinical studies have investigated 6-gingerol’s wound-healing activity due to its poor pharmacokinetic properties. However, nanotechnology can deliver 6-gingerol while possibly enhancing these properties. Our study aimed to develop a nanophytosome system loaded with 6-gingerol molecules to investigate the delivery system’s influence on wound healing and anti-cancer activities. Methods: We adopted the thin-film hydration method to synthesize nanophytosomes. We used lipids in a ratio of 70:25:5 for DOPC(dioleoyl-sn-glycero-3-phosphocholine): cholesterol: DSPE/PEG2000, respectively. We loaded the 6-gingerol molecules in a concentration of 1.67 mg/mL and achieved size reduction via the extrusion technique. We determined cytotoxicity using lung, breast, and pancreatic cancer cell lines. We performed gene expression of inflammation markers and cytokines according to international protocols. Results: The synthesized nanophytosome particle sizes were 150.16 ± 1.65, the total charge was −13.36 ± 1.266, and the polydispersity index was 0.060 ± 0.050. Transmission electron microscopy determined the synthesized particles’ spherical shape and uniform size. The encapsulation efficiency was 34.54% ± 0.035. Our biological tests showed that 6-gingerol nanophytosomes displayed selective antiproliferative activity, considerable downregulation of inflammatory markers and cytokines, and an enhanced wound-healing process. Conclusions: Our results confirm the anti-cancer activity of PEGylated nanophytosome 6-gingerol, with superior activity exhibited in accelerating wound healing.

## 1. Introduction

Human skin, the body’s largest organ, is considered the first protective barrier against environmental changes besides its other functions, such as thermoregulation and maintaining aesthetic appearance [1]. Wounded skin undergoes repairing and regenerating processes to avoid traumatized tissues and painful sensations [2]. The healing process involves the contraction of injured tissues, biosynthesizing collagen, and epithelization [3]. Untreated and unhealed wounds can develop into chronic wounds, which might lead to further complications, such as septic infection, and even the need for organ transplantation [1,4]. The internal self-healing process lasts from 5 to 10 days depending on the severity of the wound and other factors. Sometimes, the process might take up to 30 days until complete healing, with tissue hypoxia, exudation, and necrosis among the delaying factors. Furthermore, infectious wounds are very likely to develop into chronic wounds [5]. Bioactive molecules originating from natural sources have been extensively studied for their wound-healing capabilities. These include plant polyphenols, such as lupeol, curcumin, and catechin, and animal sources, such as bee venom [6].

Ginger, *Zingiber officinale*, is rich in bioactive compounds that have confirmed activities against inflammation, oxidative stress, and cancerous cells. Alongside its hypoglycemic and hypolipidemic activities, it also contributes to relieving motion sickness [1,2,3]. Its bioactive compounds include terpenes and numerous phenolic compounds, such as gingerols, shogaols, and paradols [3]. Indeed, 8- and 10-gingerol, 6-gingerol is an active principle with known therapeutic activity that is found in abundance in fresh ginger rhizomes [7].

Reports published on 6-gingerol highlight its capability in promoting the epithelization process and suppressing the production of macrophages and cytokines, thus contributing to a decrease in the inflammatory response [1,8]. This anti-inflammatory effect is advantageous because it stimulates vascularization and the wound-healing process. Moreover, in combination with vitamin D, it enhances wound healing in diabetic patients by increasing expression levels of hemopexin-like domains (HPX), fibrillogenesis (FN), and collagen, accelerating the healing process and shortening epithelialization time [9].

Various vehicles and delivery systems have been utilized to deliver therapeutic agents to their targets in optimal conditions. Nanoparticle carriers have shown promising results in treating acute and chronic wounds due to their physiochemical, optical, and biological properties. Nanocomposites are smart materials that have been synthesized by incorporating the nanoparticles into scaffolds, and they have accelerated wound-healing activity in addition to their antimicrobial, anti-inflammatory, and angiogenic properties [10,11,12].

Nanoparticles affect the healing mechanism by influencing collagen deposition and realignment, thus initiating skin regeneration. Nanotechnology is being increasingly assimilated in wound therapy because it provides flexibility, reasonable mechanical strength, and large porosity, and ensures non-adherence to the wound surface. Additionally, its cooling sensations, moistening effects, and protection from microbial contamination are desirable properties [12,13,14,15].

Buflomedil hydrochloride, a topical vasoactive agent, is an example of a therapeutic agent incorporated into nanocarriers, and it has shown improved nutritive perfusion within the lesion’s vicinity, increasing wound-healing properties [16].

Uploading high concentrations of the therapeutic agent, in addition to its direct delivery to the wounded tissue, should decrease systemic side effects; therefore, the system is a promising delivery model. Similar findings were obtained after treating calvarial bone wounds with gallic acid in the liposomal system of an animal model [17]. Quercetin-loaded liposomes exerted a sustained release and displayed effective wound-healing properties [18]. Nanoscale liposomes provide benefits when treating various illnesses, such as rheumatoid arthritis, namely reduced toxicity and enhanced delivery [13,19].

The active constituent of ginger, 6-gingerol, is known to exhibit wound healing and anti-inflammatory activities; however, it has poor pharmacokinetic properties regarding its slight solubility in aqueous media and its low bioavailability after oral administration [14]. To the best of our knowledge, there are few clinical studies on 6-gingerol activity regarding these biological effects. Therefore, in our research, we loaded 6-gingerol particles into PEGylated nanophytosomes to investigate their wound healing, anti-inflammatory, and antiproliferative properties using lung, pancreatic, and breast cancer cell lines.

## 2. Materials and Methods

### 2.1. Chemical Reagents

We purchased dried ginger rhizomes from a local market in Amman, Jordan, and we kept them at room temperature. We purchased 6-gingerol, quercetin, and gallic acid from Santa Cruz Biotechnology, CA, San Juan, USA. We purchased analytical- and HPLC-grade solvents from Sigma Aldrich, St. Louis, MO, USA, unless stated otherwise. We purchased chemicals from, Jordan Genome store, Amman, Jordan. We obtained deionized water from Millipore Waters, California, CA, USA. We obtained PANC1 pancreatic (ATCC number: CRL-1469), A549 lung (ATCC number; CCL-185), and MDA-MB-231 breast (ATCC number: HTB-26) cancer cell lines from PDRC (Pharmacological and Diagnostic Research Center at Al-Ahliyya Amman University). We purchased cell culture plates from TPP, Zollstraße, Switzerland.

### 2.2. Plant Preparation, Extraction, and Phytochemical Analysis

We crushed the dried rhizomes to reduce their size and then macerated them in methanol at a ratio of 2:3 *w*/*v*. We filtered, concentrated, and kept the mixture at 4 °C. We performed phytochemical analysis and quantification of the polyphenol and flavonoid content [15]. We tested the dried extract for its antioxidant properties by following optimized methods [20].

### 2.3. HPLC Analysis of the Ginger Crude Extracts

#### Sample Preparation

We prepared a stock solution of the authentic 6-gingerol by initially dissolving in DMSO and then diluting it with acetonitrile. We centrifuged the mixture (4000 rpm, 2 min) using a benchtop centrifuge (Eppendorf^®^, Germany). We took the supernatant and injected the 4 µL sample size into the autosampler.

### 2.4. PEGylated Nanophytosome Formation

We prepared PEGylated nanophytosomes using the thin-film hydration method (TFHM) as previously described [11,16], concentrating cholesterol (25%), 1,2-dioleoyl-sn-glycero-3-phosphocholine (DOPC) (70%), and 1, 2-distearoyl-sn-glycero-3-phosphoethanolamine-Poly (ethylene glycol) (5%) into 6-gingerol fixed at 1.67 mg/mL.

### 2.5. Characterization of PEGylated Nanophytosomes

#### 2.5.1. Size and Charge

We measured the average size, charge, and polydispersity index (PDI) for both the free and 6-gingerol-encapsulated PEGylated nanophytosomes using dynamic light scattering (DLS) on a Zetasizer Nano-ZS^®^ (Malvern Instruments Ltd., Malvern, UK). We diluted samples (1:20) with deionized water. We placed all samples in the Zetasizer specimen holder for approximately 1–2 min before measuring to equilibrate with room temperature [19].

#### 2.5.2. Encapsulation Efficiency (%EE) and Drug Loading (%DL)

We determined the encapsulated 6-gingerol’s percentage as follows. First, we degraded the loaded PEGylated nanophytosomes by adding acetonitrile at a ratio of 4:1 acetonitrile: phytosomes. Second, we centrifuged the mixture (7000, 10 min). Third, we performed the sonication step (10 min, 35 °C), then centrifuged (12,000 rpm, 10 min) and filtered with the aid of 0.45 µm syringe filters [12,19].

We measured the encapsulated 6-gingerol’s concentration using an HPLC system (UV Detector at 280 nm, HPLC Nucleodur-C18 Column; 250 mm × 4.6 mm, 5 µm), at a slow flow rate of 1 mL/min and column temperature of 40 °C using a 10 µL injection volume connected to Labsolution^®^ computer software, Weiswampach, Luxembourg. The mobile phase comprised 60:40 HPLC-grade methanol: water [14]. Equations (1) and (2) show the calculations of % encapsulation efficiency (EE) and % drug loading (DL), respectively:
(1)$$(\%\mathrm{EE})=\frac{\left[\mathrm{Encapsulated}6-\mathrm{Gingerol}\right]}{\left[\mathrm{Total}6-\mathrm{Gingerol}\right]}\times 100$$

Equation (1): The encapsulation efficiency of 6-gingerol.
(2)$$(\%\mathrm{DL})=\frac{\left[\mathrm{Weight}\mathrm{of}\mathrm{Loaded}\mathrm{drug}\right]}{\left[\mathrm{Weight}\mathrm{of}\mathrm{lipids}\right]}\times 100$$

Equation (2): The drug loading of 6-gingerol.

#### 2.5.3. Transmission Electron Microscopy (TEM)

We analyzed the structure and morphology of the empty and loaded PEGylated nanophytosomes using the TEM analysis negative staining method. First, we coated 200-mesh formvar copper grids (SPI Supplies, West Chester, PA, USA) with carbon under a low-vacuum Leica EM ACE200 glow discharge coating system (Leica, Westbahnstraße, Austria). Then, we further coated the carbon-coated grids with 1.5% Vinylec K in a chloroform solution. We placed a drop of deionized water-diluted liposome suspension on the 200-mesh copper grid, followed by an air-drying step. Then, we stained the loaded grids with a 3% v/v aqueous solution of uranyl acetate for 20 min at room temperature. After incubation, we washed the grids with distilled water and dried them at room temperature before imaging with a Versa 3D (FEI, Hoofstraat, The Netherlands) TEM operating system at an acceleration voltage of 30 kV [19].

### 2.6. Stability Study (Lyophilization of Loaded PEGylated Nanophytosomes)

We dissolved the lyoprotectant (dextrose) in phosphate-buffered saline (10% *w*/*v*). We freeze-dried PEGylated nanophytosome suspensions in a buffer with the lyoprotectant (the liquid froze at −195 °C). The dehydration step lasted for two days at −40 °C until we obtained dried powders. We stored the resulting lyophilized powders for one month and rehydrated them (when needed) to their original volume at room temperature with phosphate-buffered saline (PBS). After adding PBS, we equilibrated the samples at room temperature for 30 min. Then, we subjected the samples to the size PDI and encapsulation efficiency test [21].

### 2.7. Viability Assay

We determined cytotoxicity measurements using a 3-(4,5-dimethylthiazol-2-yl)-2,5-diphenyltetrazolium bromide (MTT) assay for cytotoxicity [22,23]. We seeded lung cancer A549, breast cancer MDA-MB-231, and pancreatic cancer Panc1 cell lines in 96-well plates at 1 × 10^4^ cells/well and cultured them in a medium containing *Zingiber officinale* extract, 6-gingerol, and PEGylated nanophytosomes loaded at concentrations of 1.5–50 µg/mL. After 48 h, we performed the MTT assay [20]. Human periodontal ligament fibroblasts (PDL) are a primary cell culture used for verifying selective cytotoxicity.

### 2.8. Gene Expression Level Assay

We plated MDA cells at a density of 1 × 10^6^ cells/well in 6-well plates [24]. After 24 h, we incubated the selected concentration of ginger extract and liposomes for another 24 h to measure the fold change in gene expressions of IL-10, IL-1beta, IL-6, IL-8, TNF-alpha, and IRAK1.

### 2.9. Ribonucleic Acid (RNA) Extraction and Analysis

We extracted total RNA using a RNeasy Mini kit^®^, QIAGEN, USA. We transferred the cell lysate to the provided spin column before centrifugation (10,000 rpm, 15 s). We performed the assay reactions following the manufacturer’s instructions. We synthesized cDNA using a reverse transcription system (Applied Biosystem, Waltham, MA, USA). We measured the optical densities at 260 and 280 nm, respectively. The ratio (A260/A280) was 1.6–1.8 for most of the cDNA-extracted samples. We performed relative-quantitative analyses of our tested genes’ mRNA expression levels using a fast SYBR green kappa master mix (Biosystem, USA). We investigated GAPDH, IL-1, beta IL-10, IL-6, IL-8, TNF-alpha, and IRAK1. We used glyceraldehyde-3-phosphate dehydrogenase (GAPDH) as an internal reference gene to normalize the expression of the tested genes [24,25].

### 2.10. Statistical Analysis

We determined statistical differences between control and treated groups using GraphPad Prism ANOVA followed by Dunnett’s post hoc test. For all statistical analyses, we considered a *p*-value of less than 0.05 as statistically significant.

## 3. Results

### 3.1. Extraction and Total Phenol, Flavonoid, and Antioxidant Contents of Crude Extract

The methanolic extraction yield was 46.32 mg/g. We measured the total flavonoids based on the quercetin standard curve as 18.52 ± 1.48 mg/g ± SD, and the total phenol content based on the gallic acid standard was 39.17 ± 1.24 SD. Furthermore, we measured the antioxidant activity of the methanolic extract compared with vitamin C in IC50 as 3.325 ± 0.21 mg/mL ± SD.

### 3.2. Effect of 6-Gingerol on Particle Size, PDI, and Zeta Potential

We formulated the free and the 6-gingerol-loaded PEGylated nanophytosomes in lipid ratios of DOPC: Cholesterol: DSPE/PEG2000 (70:25:5). Table 1 and Figure 1A show the particle sizes, PDIs, and zeta potentials of the different formulations. All the nanoliposome data were within the range of optimum formulation [25]. There were no significant differences in the average particle sizes and PDIs between the free and loaded nanoparticles; furthermore, we calculated the *p* values as 0.96 and 0.71, respectively. However, we measured significant differences in the zeta potential (charge) (*p* < 0.05), which could arise from 6-gingerol molecules’ phenolic oxygen. Accordingly, we included more dispersed negative charges [26].

Our TEM study revealed the successful loading of 6-gingerol molecules into PEGylated nanophytosomes. Our results represent the loaded nanophytosomes’ spherical shape and uniform size. We measured the average particle size as 23.46 ± 40.21 nm (*n* = 9). Figure 1A–C shows the particle size of the product and that 6-gingerol has been successfully encapsulated in the system.

#### 3.2.1. Stability after Lyophilization

We measured the loaded system’s encapsulation efficiency as 34.54 ± 0.035%. A month later, after storing the lyophilized system, we found that the retained formula’s mean of encapsulated efficiency was 33.94%.

Lyophilization should increase the nanophytosomes’ shelf-life by preserving them dry in a lyophilized cake until needed; then, they can be reconstituted in water and used. However, we observed no significant differences in our other studied factors after lyophilization, such as PDI and size, compared with data obtained before the freezing process (*p* > 0.05). We added a lyoprotectant to the final formula to maintain the particles’ distribution during the lyophilization–rehydration cycle (Table 2).

#### 3.2.2. Modulation of Proliferation of Lung and Pancreatic Cancer Cell Lines, as Well as Fibroblasts, by Ginger Extract, 6-Gingerol, and 6-Gingerol Particles Carried by PEGylated Nanophytosomes

Excluding fibroblasts, 6-gingerol-loaded PEGylated nanophytosomes exerted considerable antiproliferative efficacy against A549 and Panc1 cell lines over 48 h incubations. Furthermore, the crude extract of rhizomes and the authentic 6-gingerol sustained their antiproliferative efficacies against fibroblasts, A549, and Panc1 cell lines (Figure 2, Figure 3 and Figure 4).

#### 3.2.3. Gene Expression of Pro-Inflammatory Cytokines

We determined the gene expression levels of the cytokines in ginger crude extract and the loaded PEGylated nanophytosomes (1.5 µg/mL) with the authentic 6-gingerol using RT-PCR technology. The cytokines we examined were TNF-α, IL-1beta, IL-6, IL-8, IL-10, and IRAK1.

The breast cancer MDA-MB-231 cell line elicits the production of proinflammatory cytokines TNF-α, IL-1beta, IL-6, IL-8, and IL-10. Our analysis of IRAK1.RT-PCR expression levels revealed that the crude extract as well as the pure compound 6-gingerol have significantly (*p* < 0.05) downregulated factors (TNF-α, IL-1beta, IL-6, IL-8, and IRAK1) in MDA after being separately loaded in PEGylated nanophytosomes (Figure 5).

#### 3.2.4. Wound Healing

The effects of ginger crude extract, 6-gingerol, and PEGylated nanophytosomes loaded with 6-gingerol on the wound-healing process are shown in Figure 6. We observed a marked and considerable increase in the wound-healing process of the human periodontal ligament fibroblasts in the medicated cells compared with those left untreated.

## 4. Discussion

Inflammatory responses to harmful stimuli are among the leading causes of variable health issues in the cardiovascular and pulmonary systems, contributing to autoimmune illnesses. Furthermore, the chemicals released by inflamed and injured tissues could initiate and exaggerate the proliferation and metastasis of cancerous cells. Collectively, cytokines, which are pro-inflammatory signaling agents, enhance the survival of tumor cells through activating the NFκB pathway [27]. Considering the harmful consequences of inflammatory reactions, large quantities of research have been conducted to discover and formulate agents with anti-inflammatory potential. Ameliorating inflammation should accelerate wound-healing cascades, possibly reducing the growth of numerous cancerous cells. Several natural and synthetic agents have been investigated for their anti-inflammatory activity, and researchers have discovered that many compounds have such effects [28,29,30]. The results of previous reports have elucidated the relationship between medicinal plants and the NFκB pathway. Ginger, *Zingiber officinale*, is among the studied plants, and was approved as an alternative anti-inflammatory agent against the production of cyclooxygenase (COX)-2 and nitric oxide (NO) [31,32,33].

According to our results, we confirmed the anti-inflammatory properties of ginger crude extract and its active constituents in concomitance with previously published data. We determined this biological activity based on the expression levels of pro-inflammatory cytokines in human breast cancerous cells. This potential activity is promising for limiting breast cancer metastasis. According to our obtained data, the crude ginger extract, 6-gingerol, and 6-gingerol formulated into nanocarriers have shown that they can accelerate the wound-healing process. This activity may be due to the strong relationship between reductions in the biosynthesis of pro-inflammatory markers and enhancements in the wound-healing process [33]. Acute and chronic wounds occur after any disruption in skin integrity, either in the epidermis or dermis layers. While the skin serves as the body’s mechanical barrier to the outer environment, maintaining its integrity should protect it from infections, inflammation, and electrolyte imbalance [34]. Usually, acute wounds heal faster than chronic wounds, and we observed a delay of up to three months when comparing chronic and acute wounds. Treating chronic wounds is one of the most challenging issues facing the regenerative medicine field, and there have been extensive scientific efforts to develop therapeutic strategies in a fast, qualified, and cost-effective therapy modality [34].

A technology to utilize nanocarrier particles to deliver therapeutic agents would increase the effectiveness of treatment and revolutionize medicinal therapy. Wound-healing processes benefit from incorporating nanotechnology through minimization of the time required to achieve complete healing and improving the response rate to medicinal therapy [35]. Nanomaterials, nanoscaffolds, and nanofibers, in addition to biomaterials, are used for topical drug delivery in wound-healing processes [35]. The use of nanomaterials for biomedical and pharmaceutical applications has gained considerable attention, especially in recent years, and several nanomaterials are used in biomedical applications for wound healing, drug delivery, etc. [4,6]. We adopted a thin-film hydration method to load 6-gingerol particles because it is a simple, globally accepted, and well-optimized technique [36]. Encapsulating medicinal agents from their natural origin into nanoliposomes is a promising delivery system by which to improve overall therapeutic efficacy. For example, capsaicin loaded into nanoliposome carriers has shown enhanced stability, selectivity, and improvement in activity against human cancerous cells. The same system considerably decreased the bioproduction of IL-6 and IL-8 compared with the control [13]. Our compound of interest, 6-gingerol, has been studied in animal models for its healing activity. In these studies, 6-gingerol was incorporated in an ointment dosage form at a concentration of 10% of the total formula. Then, it was applied to wounded rats (*Rattus norvegicus*). The incision wounds showed an accelerated healing process compared with those untreated or treated with Oxyfresh Soothing Pet Gel, which was used as a positive control. The average recovery took 16 days [37]. According to Rahayu et al., applying ginger extract to incision-wounded rats reduces neutrophil concentrations in the proliferation and maturation phases [38].

Our research has produced a novel formula with remarkable stability and selectivity. Our system of PEGylated nanophytosomes loaded with 6-gingerol showed promising biological activities and good physiochemical properties. However, this study has some limitations. Flow cytometry technique should have been used and correlated with immunohistochemistry data, and dose-response at different concentrations of 6-gingerol should have been performed to determine maximum drug load ability. In addition to the MTT assay, we should have tested the effect of the compound on a macrophage cell line.

## 5. Conclusions

Our novel formulation of 6-gingerol-loaded PEGylated nanophytosomes ensured considerable stability, selectivity, and safety and offered higher wound-healing activity levels than 6-gingerol alone and the crude extract. Our results are comparable with other natural compounds using encapsulated nanoliposome technology. In the future, 6-gingerol-loaded PEGylated nanophytosomes may provide a promising approach for managing wounds and inflammatory illnesses, including cancers. Future studies would look at carrying out experiments on in-vivo wound healing and optimizing the dose of 6-gingerol loaded on the PEGylated nanophytosomes. In addition, flow cytometry will be used to study in detail the induction of apoptosis and the effect of synthesized 6-gingerol phytosomes on the cell cycle.

## Figures and Tables

**Figure 1 nutrients-14-05170-f001:**
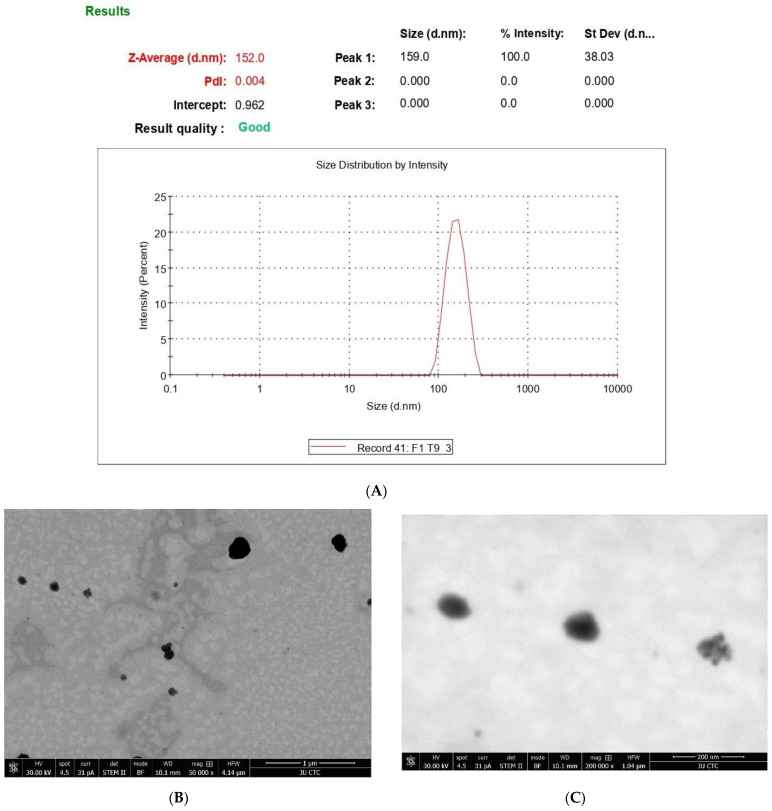
(**A**) Size distribution of 6-gingerol-loaded PEGylated nanophytosomes by DLS; (**B**) TEM images of shape and size of loaded PEGylated nanophytosomes; (**C**) morphology (6-gingerol-loaded PEGylated nanophytosomes).

**Figure 2 nutrients-14-05170-f002:**
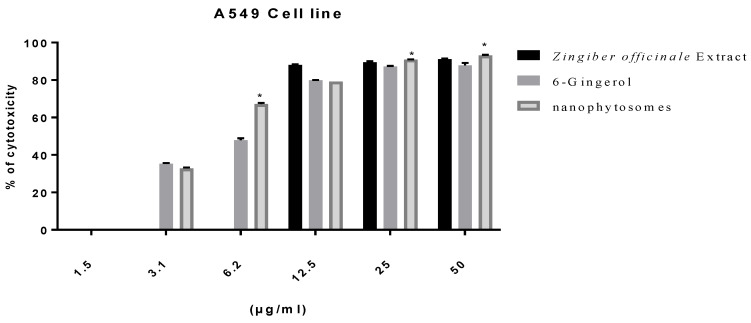
Effect of *Zingiber officinale* extract, 6-gingerol, and 6-gingerol-loaded PEGylated nanophytosomes on cytotoxicity of A549 cells compared with control (6-gingerol). * Statistically different at significance level *p* < 0.05.

**Figure 3 nutrients-14-05170-f003:**
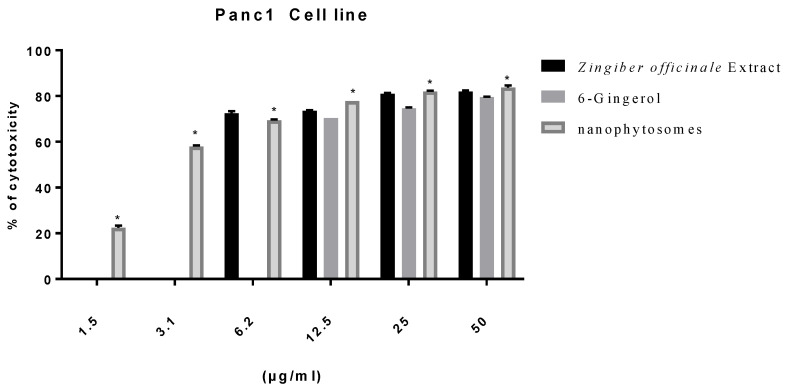
Effect of *Zingiber officinale* extract, 6-gingerol, and 6-gingerol-loaded PEGylated nanophytosomes on cytotoxicity of Panc1 cells compared with control (6-gingerol). * Statistically different at significance level *p* < 0.05.

**Figure 4 nutrients-14-05170-f004:**
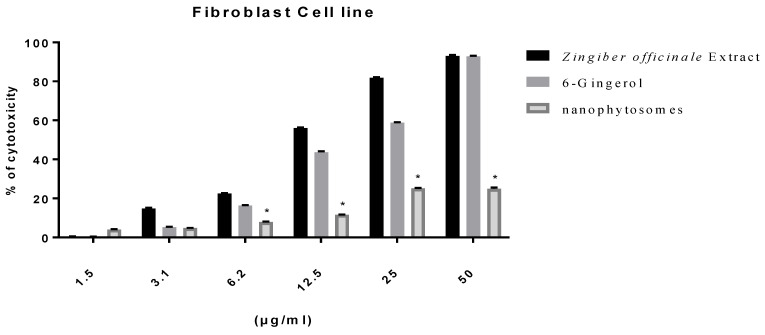
Effect of *Zingiber officinale* extract, 6-gingerol, and 6-gingerol-loaded PEGylated nanophytosomes on cytotoxicity of human periodontal ligament fibroblast cells compared with control (6-gingerol). * Statistically different at significance level *p* < 0.05.

**Figure 5 nutrients-14-05170-f005:**
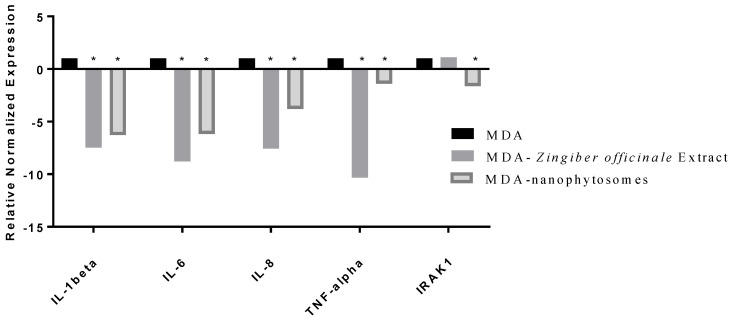
Effect of *Zingiber officinale* extract and 6-gingerol-loaded PEGylated nanophytosomes on the expression of IL-1beta, IL-6, IL-8, TNF-alpha, and IRAK1 on MDA cells compared with control (cells only). The crude extract as well as the pure compound 6-gingerol have significantly (*p* < 0.05) downregulated factors (TNF-α, IL-1beta, IL-6, IL-8, and IRAK1) in MDA after being separately loaded in PEGylated nanophytosomes. * Statistically different at significance level *p* < 0.05.

**Figure 6 nutrients-14-05170-f006:**
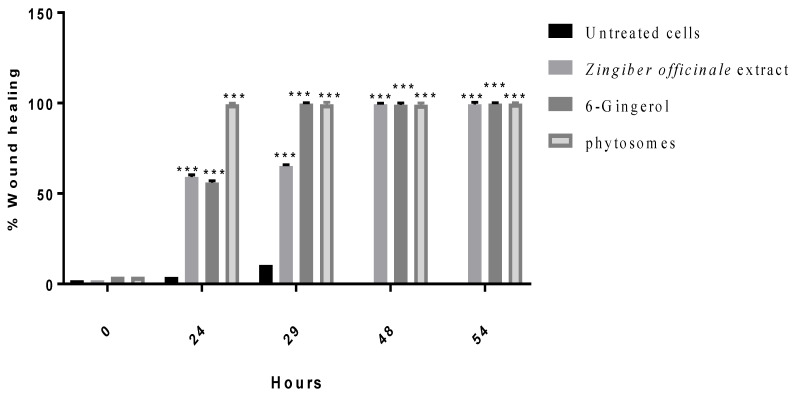
Effect of *Zingiber officinale* extract, 6-gingerol, and 6-gingerol-loaded PEGylated nanophytosomes on wound healing of human periodontal ligament fibroblasts cells compared with control (untreated cells). ***: *p* < 0.001.

**Table 1 nutrients-14-05170-t001:** Effect of 6-gingerol on nanoparticle characterization.

Description						Two-Sample t-Test
Test	Group	Mean	Std. Deviation	Minimum	Maximum	Sig.
Size	1	126.01	0.435	125.70	126.50	
	2	150.16	1.650	148.80	152.00	0.96 ^ns^
Charge	1	−11.13	0.152	−11.00	−11.30	
	2	−13.36	1.266	−12.40	−14.80	0.04 *
PDI	1	0.127	0.017	0.11	0.14	
	2	0.060	0.050	0.004	0.09	0.71 ^ns^

1: Free PEGylated nanophytosomes. 2: PEGylated nanophytosomes loaded with 6-gingerol. All data are normally distributed according to the Shapiro–Wilk normality test. *n* = 3; * Statistically different at significance level *p* < 0.05. ns: Not statistically different at significance level *p* < 0.05.

**Table 2 nutrients-14-05170-t002:** Effect of lyophilization process on characterization of PEGylated nanophytosomes loaded with 6-gingerol molecules.

Description				Two-Sample *t*-Test
Test	Group	Mean	Std. Deviation	Sig.
Size	1	150.1667	1.65025	0.102 ^ns^
	2	160.0667	9.31254	
PDI	1	0.0623	0.05054	0.628 ^ns^
	2	0.2400	0.06207	

1: The 6-gingerol-loaded PEGylated nanophytosomes before lyophilization. 2: The 6-gingerol-loaded PEGylated nanophytosomes after lyophilization. All data are normally distributed according to the Shapiro–Wilk normality test, *n* = 3; ns: Not statistically different at significance level *p* < 0.05.

## Data Availability

Study data are available from the authors upon request.

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
