# Peer review of "Preparation, Characterization, Wound Healing, and Cytotoxicity Assay of PEGylated Nanophytosomes Loaded with 6-Gingerol"

_nutrients, 2022, doi:10.3390/nu14235170_

Round 1

Reviewer 1 Report

The manuscript entitled “Preparation, Characterization, Wound healing, and cytotoxicity assay of PEGylated Nano-Phytosome loaded with 6-Gingerol” falls within the scope of the Journal. However, this reviewer has the following comments for the manuscript.

Major comments:

- Authors should rewrite the introduction section, which needs to be shorter and focused on the study's objectives.

- Authors should report the statistical analysis in the figure legends.

- Authors reported, “Lung cancer A549 cell line, breast cancer MDA cell line and pancreatic cancer Panc1 Cell lines, were seeded in 96-well plates at 1×104 cells/well and cultured in the medium containing Zingiber officinale extract, 6-gingerol and PEGylated - Nanophytosomes loaded at concentrations (1.5-50 µg/ml).” Authors should explain the choice of these cell lines. In addition, authors should also evaluate cytotoxicity at 4 and 24 hours.

- Authors reported “The levels of gene expression of the cytokines in ginger crude extract and the loaded PEGylated-Nanophytosomes (1.5µg/ml) with the authentic 6-gingerol were determined by RT-PCR technology. The examined cytokines were TNF-α, IL-1beta, IL-6, IL-8 285 and IL-10 and the IRAK1” Because the anti-inflammatory activity is also evaluated, authors should test the effect of the compound on a macrophage cell line (RAW or J774), performing MTT  assay on this cell line.

Minor comments:

- Authors should insert a graphical abstract that summarizes the contents of the article in a concise form in order to capture the attention of the readership.

- Authors should report the keywords in alphabetical order.

- Authors should insert an abbreviation section. The words for which is specified an abbreviation should be written in full the first time they are mentioned.

- The English language has to be extensively revised.

- Authors should improve the formal aspects of the manuscript.

- Authors should improve the resolution of all Figures.

Author Response

We would like to thank all reviewers very much for giving us time to revise our work. Your comments were very valuable to us and should contribute to improving the current manuscript. We hope that the corrections we made meet your approval. 

Reviewer 2 Report

The authors carried out an interesting study on the synthesis of nanophytosomes containing encapsulated 6-gingerol, at the same time, studies were carried out with ginger extract, which is widely used in traditional medicine. It has been shown that this approach makes it possible to achieve good solubility of 6-gingerol in an aqueous medium and to study its antitumor properties on cell lines.

On the one hand, the article is interesting and may be useful to researchers, meanwhile, there are doubts about the validity of the study of the properties of ginger extracts, since they cannot be standardized for subsequent use in the pharmacopeia. In addition, it is known that the study of cytotoxicity using the MTT approach has recently been strongly criticized due to the large measurement error and the presence of a large number of limitations. In this study, it is highly desirable to study the state of cells using modern methods of flow cytometry and study in detail the induction of apoptosis and the effect of synthesized phytosomes on the cell cycle.

Is there an error in calculating the polydispersity index (0.060 ± 0.050)?

Author Response

(The authors gave the same response as above.)

Reviewer 3 Report

The manuscript entitled “Preparation, Characterization, Wound healing, and cytotoxicity assay of PEGylated Nano-Phytosome loaded with 6-Gingerol” aims to develop a nanophytosome system loaded with 6-gingerol molecules and to investigate the influence of the delivery system on wound healing and anti-cancer activities on lung, breast, and pancreatic cancer cells. The obtained results revealed anti-cancer activity of PEGylated-Nanophytosome 6-gingerol with a superior activity in accelerating wound healing.

Page 2, line 53 – in the sentence “Bioactive molecules isolated from plant extracts have been extensively studied for their wound healing capabilities” it could be included from plant extract and animals since some constituents from venoms have similar potential (e.g. bee venom has wound healing properties as well)

Page 2, lines 76 to 86 - would fit better in the Discussion section

In the last paragraph of the Introduction section I would suggest to explain the aim of the study in more details. Briefly mention cell lines used in the study.

Page 5, line 193 – please write breast cancer MDA-MB-231 cell line and so on…

Figure 1 – are there a better resolution images for Fig. 1?

Minor remarks:

Put all Latin words in italic (e.g. in vitro, etc)

Pay attention on the unit writing in the Figures (e.g. ml vs mL) in line with Authors guidelines.

Author Response

(The authors gave the same response as above.)

Round 2

Reviewer 2 Report

The authors made a significant edit to the text of the manuscript.